# Early Transport Patterns and Influencing Factors of Different Stocks of *Uroteuthis edulis* in the East China Sea

**DOI:** 10.3390/ani14060941

**Published:** 2024-03-19

**Authors:** Nan Li, Qinwang Xing, Zhiping Feng, Xinjun Chen, Zhou Fang

**Affiliations:** 1College of Marine Living Resource Sciences and Management, Shanghai Ocean University, Shanghai 201306, China; d210200060@st.shou.edu.cn (N.L.); fzp43256@163.com (Z.F.); xjchen@shou.edu.cn (X.C.); 2Institute of Marine Science and Technology, Shandong University, Qingdao 266237, China; qinwangxing@gmail.com; 3National Engineering Research Center for Oceanic Fisheries, Shanghai Ocean University, Shanghai 201306, China; 4Key Laboratory of Sustainable Exploitation of Oceanic Fisheries Resources, Ministry of Education, Shanghai Ocean University, Shanghai 201306, China; 5Key Laboratory of Oceanic Fisheries Exploration, Ministry of Agriculture and Rural Affairs, Shanghai 201306, China; 6Scientific Observation and Experimental Station of Oceanic Fishery Resources, Ministry of Agriculture and Rural Affairs, Shanghai 201306, China

**Keywords:** East China Sea, *Uroteuthis edulis*, trace elements, particle tracer, migration

## Abstract

**Simple Summary:**

The life history traits of *Uroteuthis edulis*, an economically important loliginid squid in the East China Sea (ECS), have always attracted attention, especially its migration. Understanding the early transport process and influencing factors is a prerequisite for grasping population recruitment and dynamics. To address this, this study used statolith microchemical information and particle tracer experiments to indicate changes in individual habitats and simulate early transport processes, respectively. The results showed that there were differences in early transport trajectories between spring stock and summer stock. Velocity, salinity, and temperature were the key environmental variables affecting early transport for spring stock, while mixed-layer depth (MLD), velocity, and chlorophyll a concentration (Chla) were the key environmental variables for summer stock. These results are related to the dominant characteristics of ocean currents during early transport, which in turn affects the growth and development status of paralarvae. This study provides scientific guidance for understanding population dynamic processes and the sustainable development of resources.

**Abstract:**

*Uroteuthis edulis* (*U. edulis*) is an important economic loliginid resource in the East China Sea (ECS). Its flexible life history traits enable the population to quickly adapt to changes in habitat. Understanding the early transport process helps us to grasp the habitat requirements of populations at key life history stages. In this study, particle tracing was used to simulate the early transport trajectories (within 120 days). The gradient forest method (GFM) and generalized additive mixed models (GAMMs) were used to analyze the key environmental variables that affect the early transport trajectories and the impact of environmental factors on the transport process, respectively. The results showed that spring stock tracers were transported to the northeast of the release area (Pengjiayu water) and the Pacific side of Japan. Summer stock tracers were transported to the north and northeast of the release area (Zhoushan island). Current velocity, salinity, and temperature were key environmental variables that affected the trace element ratios of spring stock at early life history stages. Mixed-layer depth (MLD), velocity, and chlorophyll a concentration (Chla) were key environmental variables for summer stock. Zonal velocity was positively correlated with the trace element ratio for spring and summer stock (0.14–0.16 m/s), while the meridional velocity showed an opposite correlation. The physical driving mechanisms of the Kuroshio warm current (or the Taiwan warm current) and the Yangtze River determine the paralarva retention location during early transportation. The differences in the dominant factors of the water environment in the retention area may affect the paralarva physiological functions and food availability. This study provides a scientific basis for a comprehensive understanding of the migration characteristics of *U. edulis* with different stocks.

## 1. Introduction

During the life cycle of marine organisms, individuals undergo a transition from the passive drifting of paralarva to the active long-distance migration of adults between feeding and spawning grounds [1]. The environment of spawning grounds and paralarva transport routes regulate the reproductive success of adults, which determines the dynamic growth, migration, and recruitment mechanisms of the organisms [2,3]. Therefore, understanding the early transport process of marine organisms can help to understand the population dynamics and habitat requirements at key life history stages [4].

During early transport, environmental variation (temperature, salinity, etc.) and food availability affect individual recruitment [5,6,7]. Generally, temperature affects paralarva body size by affecting hatching cycles and yolk utilization rates [6]. In addition, fluctuations in plankton biomass affect paralarva growth and mortality in nursing grounds (especially in upwelling areas), which determines the survival status of paralarva [8,9]. Therefore, changes in the physicochemical characteristics of environmental factors and food availability in early habitats are particularly important for individual growth and survival [10,11].

Offshore bottom fishery resources are gradually decreasing, while other mid-pelagic fishery species are attracting increasing attention [12]. *Uroteuthis edulis* is a short-lived cephalopod widely distributed in the coastal waters of the northwestern Pacific Ocean, especially in the East China Sea (ECS), with the annual catch being 1.5 × 10^4^ tons [13,14,15]. Several ocean currents converge in the ECS to form a complex and diverse marine environment, providing sufficient materials for the growth of plankton [16,17]. This species shows rapid generational renewal, and four stocks are determined according to the hatching characteristics throughout the year [18,19]. Among them, spring stock (squid hatched from March to May) and summer stock (squid hatched from June to August) are the main fishing groups in autumn and winter in the north waters of the ECS [4,19]. Throughout the life cycle, spring stock hatches in the waters of Pengjiayu in the northeast of Taiwan [20]. The migration route is related to the changes in the Kuroshio current and the Taiwan warm current, and is significantly affected by the monsoon [4,21]. The summer stock hatches in the waters of Zhoushan, Zhejiang, and paralarvae migration route is affected by the Yangtze River and Taiwan warm current [13,22]. In addition, in the feeding ground, spring stock body size is larger than that of summer stock, which is related to the habitat and prey abundance experienced by the stock, especially in its early life history [19,20].

The calcified tissues of Cephalopods (statolith, beaks, etc.) have unique temporal and spatial inversion properties, and the trace element signatures deposited are potential natural markers for population traceability [23,24,25]. The differences in the ratios of trace elements in statoliths (Mg:Ca, Sr:Ca, and Ba:Ca) are used to determine the population structure and predict the adult habitat based on the negative correlation between Sr:Ca and the thermal environment [1,26]. In addition, with the improvement of population dynamics research, the physical coupling model has gradually become a new way to explore the early transport of populations [7,27]. Therefore, the chemical markers of calcified tissues of Cephalopods are effective indicators for revealing life history traits, and physical models will help us to grasp the early transport process.

*U. edulis* have a wide migration range in the East China Sea, and the life history traits of different stocks are closely related to changes in habitat [15,22]. In the early life cycle (within 120 d), water temperature and current velocity are key environmental factors that affect the daily growth of spring stock, while water temperature and MLD significantly affect summer stock [19]. In addition, the dynamic growth process of squid is highly regulated by biotic and abiotic factors in the habitat environment, and the relationship between changes in environmental factors and early migration routes will be the key to analyzing early life history traits [28,29].

In this study, particle tracer experiments were first used to simulate the early transport trajectory. Secondly, assuming that the statolith trace elements can represent changes in the habitats of squid [4,22], the gradient forest method (GFM) was used to determine the key environmental variables affecting early transport. Finally, a single generalized additive mixed model (GAMM) of element ratios and environmental variables was established to analyze the impact of the environment on the transport process. This study can help better understand the early migration processes of *U. edulis*.

## 2. Materials and Methods

### 2.1. Sampling and Measurements

Samples were randomly collected monthly by a commercial trawler vessel, Zhelingyu 23860, in the ECS (124° E to 127.5° E, 29° N to 31.5° N) in three periods: from September 2018 to January 2019, September 2019 to January 2020, and September 2020 to January 2021 (Figure 1). Samples’ biological data were measured and obtained in the laboratory (mantle length (ML), body weight (BW), sexual maturity, etc.) [4]. From 2018 to 2020, 290, 224, and 259 samples were respectively collected (Appendix A).

### 2.2. Environmental Data

The spatial and temporal distributions of environmental factors and seasonal changes affect the life history traits of squids in the ECS [30]. Generally, temperature is one of the important abiotic environmental factors that triggers species reproduction and growth characteristics [20,27]. Salinity is largely determined by the coastal currents in continental shelf seas and affects the osmotic pressure balance of squid [31,32]. The chlorophyll a concentration (Chla) and mixed-layer depth (MLD) affect the nutrient concentration and average light intensity at different water layers, which in turn affect the distribution and abundance of prey [6,19]. The current velocity strength is affected by the residence time of paralarvae during the early migration process [28,30]. Based on the results of Li et al. [4], this study assumed that the spring stock and summer stock of *U. edulis* inhabit the 55 m and 25 m water layers during the entire life cycle, respectively. Therefore, the temperature at depths of 25 m and 55 m (T25, T55), salinity at depths of 25 m and 55 m (S25, S55), zonal velocity at depths of 25 m and 55 m (ZV25, ZV25), meridional velocity at depths of 25 m and 55 m (VV25, VV25), MLD, and Chla were selected to analyze environmental factors affecting the early migration of squid. All environmental data were downloaded from the website of the US National Oceanic and Atmospheric Administration (http://apdrc.soest.hawaii.edu/las/v6, accessed on 21 August 2023). The spatial resolution of all downloaded environmental variables was processed as 0.25° × 0.25° by Matlab software (Version 9.4.0.813654). The spatial and temporal ranges of environmental variables were determined with sample age and habitat location, respectively.

### 2.3. Measurements of Statolith Trace Elements

Statolith preparation and age counting were carried out by using the method of Li et al. [19], and population structure was determined according to the rule of “one day-one increment” [33]. In this study, based on the hatching time, ML, and gonad maturity composition of the samples in the sampling year, 44 and 25 polished statoliths were selected from the spring stock and summer stock for trace element determination, respectively (Table 1).

The concentrations of trace elements in statoliths were measured via laser ablation inductively coupled plasma mass spectrometry (LA-ICP-MS) in the Key Laboratory of Sustainable Development of Fishery Resources of the Ministry of Education, Shanghai Ocean University. The statolith preprocessing, instrument parameters, and calibration-standard samples were based on the method of Li et al. [4,13,22]. Statolith trace elements were measured via consecutive laser ablation (diameter of 40 μm) from the core to the edge, with an interval of 60 μm (ablation core). If the distance between the last ablation points and the core was not an integral multiple of 60 μm, the trace element was measured at the statolith edge (Figure 2) [4]. The extraction of element concentration was completed through ICPMS-Data-Cal software v.1 [22,34].

### 2.4. Tracer Experiments

Li et al. [4] discovered that the spawning grounds of spring stock and summer stock were located in the northeastern waters of Taiwan and the waters of Zhoushan, Zhejiang, respectively. In order to verify the early migratory transport process (within 120 days), this study conducted tracer experiments in the above-mentioned areas (Appendix A, areas A and B), and the tracers of the spring stock and summer stock were set to be released in the water layers of 55 m and 25 m, respectively [4]. The tracer release locations of spring stock (Appendix A, area A) and summer stock (Appendix A, area B) were 122° E to 123° E, 26° N to 27° N and 122.5° E to 123.5° E, 29° N to 30° N, respectively. In addition, tracer experiments were conducted for each monthly group of spring stock (March to May group) and summer stock (June to July group) in the sampling year, and the monthly groups released tracers in the first ten days (5th), middle ten days (15th), and last ten days (25th) to fully simulate the transport trajectory (within 120 days). The method of Xing et al. [7,35] was used to calculate the tracer advection and diffusion, and to draw their transport trajectories. Since the *U. edulis* had a diurnal vertical movement phenomenon, and the squid inhabited deeper water layers during the day, the tracer release time was 5:00 a.m. to 5:00 p.m. daily (Beijing) [17,36].

### 2.5. Statistical Analysis

According to the minimum detection limit (LOD) and relative standard deviation (%RSD) of the statolith trace elements in the standard sample, Na, Mg, Sr, and Ba were selected as the effective elements [4,22]. All data analyses were expressed as ratios of effective elements to calcium (element ratio). The results of Li et al. [22] showed (Appendix A) that the distance from the statolith core to the trace element sampling point core was divided into five regions, which corresponded to the life history stages of the squid. An analysis of the relationship between the cumulative daily incremental width and age showed that the cumulative daily increment width (≤360 μm) within 120 days corresponds to the embryonic stage until the sub-adult stage (Appendix A). Therefore, 0–120 days were selected as the early life history stages to analyze environmental factors affecting the early migration of *U. edulis*.

For spring stock (Figure 2a), the embryonic–larval stages (S1–S2), juvenile stage (S3), and sub-adult stage (S4) correspond to 3 (0 μm, 60 μm, 120 μm), 2 (180 μm, 240 μm), and 2 (300 μm, 360 μm) trace element sampling points, respectively [4,22]. According to the migratory characteristics, the habitats of spring stock were divided into three regions at early life history stages, and it was assumed that all of the monthly groups experienced the same habitats with 120 days (Figure 3). With the same research reasoning for summer stock (Figure 2b and Figure 3), the embryonic–larval stages (S1–S2), juvenile stage (S3), and sub-adult stage (S4) correspond to 3 (0 μm, 60 μm, 120 μm), 1 (180 μm), and 3 (240 μm, 300 μm, 360 μm) sampling points, respectively. In addition, since the time span of laser ablation diameter (40 μm) covered age information, this study selected the age ranges corresponding to the sampling points at the embryonic–larval stages, juvenile stage, and subadult stage to be 10 d, 10 d, and 20 d, respectively (Appendix A). Therefore, the temporal and spatial ranges of environmental variables corresponding to the statolith trace element sampling points were determined by the ablation age and potential habitat, respectively (Figure 2 and Figure 3).

With multiple potentially correlated variables as predictors, the gradient forest method (GFM) can provide the goodness-of-fit to the response variables and the weight of each predictor, as well as enable the analysis of the factors and threshold ranges of biological traits [37]. Therefore, the GFM was used to analyze the importance of environmental variables in affecting changes in element ratios at early life history stages. The average statolith trace element ratio was used as the response variable, and environmental variables corresponding to the sampling point were used as the predictor variable. In order to further clarify the nonlinear relationship between transport processes and environmental variables as well as the randomness of the samples in different sampling years, generalized additive mixed models (GAMMs) were used to analyze the quantitative relationship between element ratios and environmental variables at early life history stages [38]. To avoid collinearity among predictor variables, the response variables were modeled with a single predictor variable, and the optimal model was selected based on the maximum R^2^ and the minimum Bayesian information criterion (BIC) [38,39]. The model was written as follows:Log(STE) = S(PV1_t_) + Random(f_r1_ = ~1) + ε_t_,(1)
where STE was the response variable representing the element ratio (Na:Ca, Mg:Ca, Sr:Ca, and Ba:Ca); PV1_t_ was the predictor variable representing environmental variables corresponding to the sampling point; S() was nonlinear function; ε_t_ was a random error; and f_r1_ was a random variable (the sampling year was used as a random variable). The charts in this paper were drawn with Microsoft Excel 2019 software. The GFM and GAMMs were carried out with the “GradientForest” (Version 0.1-17) and “mgcv” (Version 1.8-31) packages in the R language (Version 3.6.3), respectively. The tracer experiments were implemented in Matlab, following the methods of previous studies [7,35].

## 3. Results

### 3.1. Tracer Trajectories

The tracer trajectories of the March to May group (spring stock) released in area “A” were similar (Figure 4, Appendix A). Within 120 d, tracers were mainly transported in two directions: to the northeastern sea area of the release area (central and northern ECS) and to the Pacific side of Japan. The tracer trajectories of the June to July group (summer stock) released in area “B” were similar (Figure 5 and Appendix A). Within 120 d, tracers were mainly transported in two directions: to the north of the release area and to the northeastern area.

### 3.2. Environmental Variable Importance

In the early life cycle (within 120 d), T55 and S25 were the most important environment variables affecting the statolith Na:Ca of spring stock and summer stock, respectively (Figure 6); VV55 and Chla were the most important environment variables affecting the statolith Mg:Ca of spring stock and summer stock, respectively (Figure 6); VV55 and MLD were the most important environment variables affecting the statolith Sr:Ca of spring stock and summer stock, respectively (Figure 6); and S55 and VV25 were the most important environment variables affecting the statolith Ba:Ca of spring stock and summer stock, respectively (Figure 6). Therefore, velocity, salinity, and temperature were the key environmental variables affecting early transport for spring stock, while MLD, velocity, and Chla were the key environmental variables for summer stock.

### 3.3. Relationship between the Element Ratio and Environmental Variables

In the early life cycle of spring stock (Table 2, Figure 7), T55 showed a negative correlation with element ratio (20–23 °C); S55 showed a positive correlation with element ratio (33.5–33.9‰); ZV55 showed a positive correlation with element ratio (0.14–0.16 m/s); VV55 showed a negative correlation with element ratio (0.08–0.20 m/s); MLD showed a negative correlation with Na:Ca (20–50 m), MLD showed a positive correlation with Mg:Ca, Sr:Ca, and Ba:Ca (20–50 m); and Chla showed a positive correlation with element ratio (0–5 mg/m^3^). For summer stock (Table 3, Figure 8), T25 showed a negative correlation with Sr:Ca (20–28 °C), T25 showed a positive correlation with Na:Ca, Mg:Ca, and Ba:Ca (20–28 °C); S25 showed a positive correlation with element ratio (32.9–33.3‰); ZV25 and VV25 showed a positive correlation with element ratio (0.05–0.15 m/s); MLD showed a negative correlation with element ratio (20–60 m); Chla showed a negative correlation with Na:Ca (0–3 mg/m^3^); and Chla showed a positive correlation with Mg:Ca, Sr:Ca, and Ba:Ca (1–3 mg/m^3^).

## 4. Discussion

### 4.1. Early Transport Trajectory

The ECS includes current systems such as the Kuroshio warm current, Taiwan warm current, and coastal currents, and their seasonal changes directly affect the hydrological conditions and economic fishing locations of the ecosystem [16,40]. In addition, the runoff of the Yangtze River and nutrient offshore expansion also affect the distribution of primary productivity in the summer [2]; therefore, the physical mechanisms of water mass and seasonal changes in prey abundance affect the feeding environments and living conditions of species in the ECS [12,41].

As a coastal loliginid squid, *U. edulis* has a flexible life history and responds quickly to changes in habitat [12,20]. The spawning grounds of the spring stock and summer stock are located in the Pengjiayu waters in northeastern Taiwan and the Zhoushan islands, Zhejiang, respectively [4]. Among them, in the Pengjiayu waters, a cyclonic upwelling cold eddy exists all year round, which causes the nutrients to be sufficiently mixed in the area [8,17]. The Zhoushan waters have natural reef landforms and nutrients that allow populations to spawn and nurse [13]. The tracer experiments showed that one branch of the spring stock was transported to the northeastern waters of the ECS with the Taiwan warm current and Kuroshio warm current, and another branch to the Pacific side of Japan with the Kuroshio current (Figure 3 and Appendix A). The summer stock was transported to the north and northeast of Zhoushan island with the Yangtze River runoff and Taiwan warm current, respectively (Figure 4 and Appendix A). Compared with the other stock, the winter stock in the Sea of Japan and the squid caught on the Pacific side of Japan are all hatched in the southern waters of the ECS [36]. The paralarvae move along the eastern continental shelf of the ECS to the Sea of Amakusa with the Kuroshio warm current in the early life cycle [27,28]. Due to the lower temperature in the Sea of Amakusa, the paralarva development and the time of entering the Tsushima Strait are delayed, which lead to the difference in the body sizes of squid on both sides of Japan [42]. The early transport trajectory of the spring stock in the ECS is similar to the winter stock in the Sea of Japan [28]. The seasonal strength relationship between the warm current and coastal current in the ECS may result in differences in the migration routes of captured adults in the ECS and the Sea of Japan [8,42]. Moreover, the mechanism of the difference in catch and biological characteristics among fishing areas still needs to be further explored [43].

For the summer stock, the Yangtze River runoff and the Taiwan warm current are important currents that affect early transport, and these currents’ interannual strength variations determine the transport distance [4,16]. The interannual variations in the Yangtze River runoff in summer are related to the input of terrestrial freshwater and nutrient concentration, and the diffusion affects the trophic levels and habitats of paralarvae [2,6,15]. In summer, the Taiwan warm current mainly originates from the Taiwan Strait water and the surface water (subsurface) of the Kuroshio warm current, which exhibits the characteristics of high temperature and low salinity [8,44]. Moreover, the interannual variations in the Kuroshio intrusion paths and the water flow in the Taiwan Strait jointly affect the distribution and physicochemical characteristics of Taiwan warm current (inshore branch) at different water layers [8]; therefore, the physical driving sources of the Yangtze River runoff and the Taiwan warm current (inshore branch) determine the transport direction of the summer stock, and the dynamic interannual variations in each component current system affect the transport distance.

The establishment of biological migration models has always been the focus in the exploration of individual migration routes and fishing ground formation mechanisms [42,43,45]. Tracer experiments can trace the transport trajectories of floating fish eggs or planktonic larvae, and the differences in the selection of tracer release areas may affect the real migration process for each stock [46]. During the early migration process, the growth rate, retention rate, and survival rate of paralarva in the ocean currents determine the resource recruitment, and food availability as well as energy efficiency affect the success rate [7,47]. Therefore, in the future, on the basis of fully determining the spawning grounds of *U. edulis*, a physical ocean model coupled with biological factors will be used to simulate the dynamic transport trajectory.

### 4.2. Relationship between Trace Element and Environmental Variables

There is a complex dynamic process in the deposition of trace elements in statoliths, and the incorporation can be regulated by intrinsic factors (growth, maturity, feeding, etc.) and extrinsic factors (salinity, temperature, water chemistry, etc.) [48,49]. The element ratio is a natural label indicating changes in habitat by species during developmental migration [50]. At the early life history stages, variations in the element ratio are related to yolk nutritional and habitat water environments [1,5]. The spring stock hatches in the Pengjiayu waters, and then paralarvae passively migrate along the Kuroshio warm current and Taiwan warm current [4,20]. The Kuroshio warm current exhibits high temperature and high salinity characteristics all year round, and paralarvae grow and develop during the transport process of the zonal current [8,16]. At this time, the thermohaline characteristics of water environments may affect the element precipitation by regulating the physiological functions, which make velocity, salinity, and temperature important environmental variables (Figure 6) [19,22]. The summer stock hatches in the Zhoushan waters, and then the paralarvae passively migrate along the Yangtze River runoff and branches of the Taiwan warm current [4,15]. In summer, the Yangtze River transports a large amount of eutrophic water, forming a nutrient zone extending to Jeju Island [2,6]. In addition, the increase in temperature leads to water stratification in the summer, which affects the nutrient supply based on horizontal advection at different layers [31]. Due to the rich nutrients carried by the Yangtze River runoff in summer and the shallowing of the mixed layer, food availability is superior at the early life history stages for the summer stock [16,19]. At this time, the nutritional level may affect the element precipitation by regulating paralarva feeding conditions, which make MLD, velocity, and Chla important environmental variables (Figure 6) [4,22]. Therefore, currents transport paralarvae to specific areas for growth and development, and the physicochemical characteristics as well as nutritional levels in water environments affect squid physiological functions [4,27,43]. The differences in dominant ocean current characteristics may influence the importance of environmental variables for trace element ratios [22].

At present, although tank experiments provide limited understanding of the quantitative relationship between water environments and statolith trace elements for *U. edulis*, reasonable analyses of physiological function regulation based on abiotic factors will help understanding [4,29]. The velocity and direction of ocean currents are the key to determining paralarvae’s growth, in which water chemical properties and prey abundance affect the survival status [43]. Salinity and temperature maintain a dynamic balance of yolk utilization rates in hatchlings, and they regulate the paralarva biochemical reaction rates and osmotic pressure, respectively [10]. MLD and Chla affect the nutrient concentration and average light intensity at different water layers through marine physical and biological processes, thereby affecting the food availability of predators [6,19]; however, *U. edulis* is tolerant to changes in the physical and chemical properties of its habitat [18,21]. Throughout its life cycle, the water temperature and salinity in the habitat are 12–27 °C and 32–34.7‰, respectively, of which 15–20 °C is more suitable for hatching and developing [19,28]. Based on the above inference, the spring stock grows and develops in the zonal advection transport of the Kuroshio warm current and Taiwan warm current (high temperature characteristic) [8,30]; however, the increase in meridional velocity may prove to be an obstacle to the time and survival state for paralarvae to enter suitable habitats [19]. At the same time, the inhabiting temperature of the spring stock gradually increases at the early life history stages, and the thermal environment may exceed the tolerance range of paralarvae [28,29]. These factors affect the element exchange between blood and lymph in the statolith, which may cause the elemental ratio to be negatively correlated with water temperature and meridional velocity for spring stock (Figure 7) [4,29]. In addition, the summer stock grows and develops in the transport of the Yangtze River runoff and Taiwan warm current (nutrient-dominated), and the nutrition concentration carried in the advection decreases with an increase in transport distance [2,4]. At the same time, in summer, the nutrient stratification phenomenon is obvious, and the food availability in the deeper mixed layer is reduced [31]. These factors affect the element absorption from prey nutrients by the lymph fluid in the statolith, which may cause the elemental ratio to be negatively correlated with velocity and MLD for the summer stock (Figure 8) [13,19]. Therefore, the environmental response difference of the statolith element ratio between the spring stock and summer stock is related to the dominant water environment characteristics in the early transport currents [8,22,28]. The hydrodynamic and physiological mechanisms of the trace element absorption still need to be further explored.

### 4.3. Early Potential Transport Process

The ECS is one of the western shelf ecosystems in the northwest Pacific Ocean [12]. Seasonal (or interannual) changes in biotic and abiotic factors directly affect the abundance and composition of communities in the ECS [16]. As short-lived species, the recruitment of loliginids is highly regulated by habitat [10]. Among them, the habitat environment in the early life history is crucial to individual growth, especially the transition from paralarvae to the larvae stage, which is closely related to the physicochemical properties and nutritional levels of ocean currents during the early transportation [19,43].

Based on the above analysis, this study analyzed the early transport process of *U. edulis*: The paralarvae of the spring stock were passively transported along the Kuroshio warm current and Taiwan warm current. The physical and chemical characteristics of key water environment variables in the Kuroshio warm current (velocity, salinity, and temperature) affected the biochemical reaction rate [4,8,28]. The paralarvae of the summer stock were passively transported along the Yangtze River runoff and Taiwan warm current (branch). The food availability of the water environment in the Yangtze River runoff affected the feeding nutritional condition, which in turn determined the growth status of squid [4,15,19]. Therefore, the physical driving mechanism of the Kuroshio warm current (or the Taiwan warm current) and the Yangtze River runoff determine the paralarva retention location during early transportation [4,43]. The differences in dominant factors of the water environment in the retention area may affect the paralarva physiological functions and food availability, which may be an important reason for the differences in the biological characteristics of different stocks [19,20,21].

## 5. Conclusions

The tracer experiments within 120 d showed that one branch of the spring stock was transported to the northeastern waters of the ECS with the Taiwan warm current and Kuroshio warm current, and another branch to the Pacific side of Japan with the Kuroshio current. The summer stock was transported to the north and northeast of Zhoushan island with the Yangtze River runoff and Taiwan warm current, respectively. The physicochemical characteristics and nutritional levels of the water environment in ocean currents affected the early transport process of *U. edulis* by regulating physiological functions. Therefore, the physical driving mechanism of the Kuroshio warm current (or the Taiwan warm current) and the Yangtze River runoff determined the paralarva retention location during the early transportation. The differences in the dominant factors of the water environment in the retention area might affect the paralarva physiological functions and food availability, which might be an important reason for the differences in the biological characteristics of different stocks.

## Figures and Tables

**Figure 1 animals-14-00941-f001:**
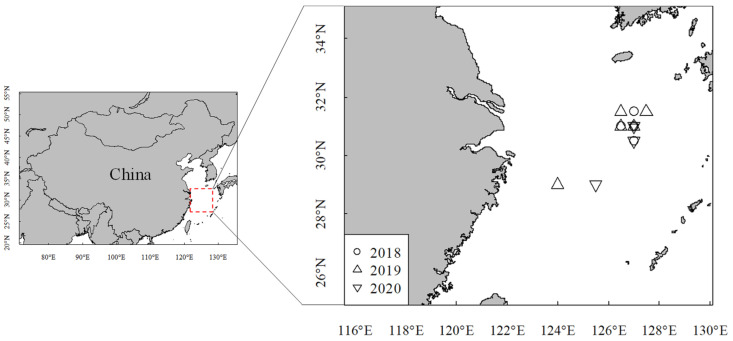
Sampling stations of *U. edulis* in the East China Sea. The sampling years are represented by 2018 (September 2018 to January 2019), 2019 (September 2019 to January 2020), and 2020 (September 2020 to December 2020).

**Figure 2 animals-14-00941-f002:**
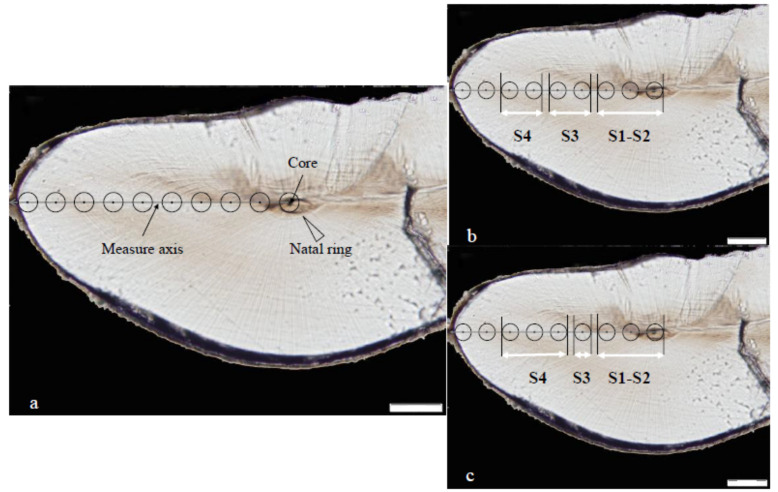
Analytic spots in statoliths of *U. edulis* (distance from core to edge: 547 μm, age: 220 d). (**a**) The black circle represents the sampling point (40 μm in diameter). The gray line shows the analytic spot direction. (**b**) The trace element sampling points corresponding to early growth stages for spring stock from the embryonic stage to the sub-adult stage. (**c**) The trace element sampling points corresponding to early growth stages for summer stock from the embryonic stage to the sub-adult stage. S1–S2, S3, and S4 represent the embryonic–larval stages, juvenile stage, and sub-adult stage (S4), respectively.

**Figure 3 animals-14-00941-f003:**
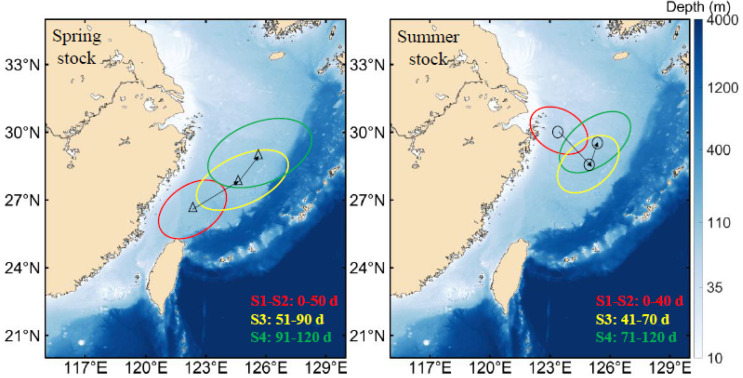
Potential distribution area of the spring stock and summer stock at early growth stages. The color circles (red, yellow, and green) and arrow show potential habitats and migration direction, respectively. S1–S2, S3, and S4 represent the embryonic–larval, juvenile, and sub-adult stages, respectively.

**Figure 4 animals-14-00941-f004:**
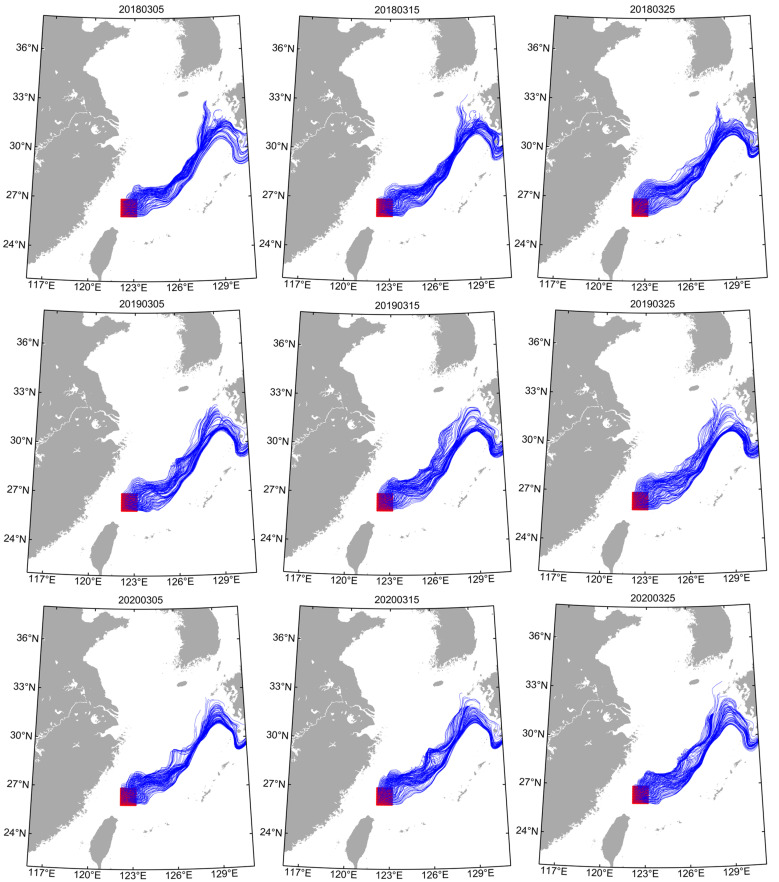
Particle tracking trajectories of the March group (spring stock) for 120 days in different sampling years. Red areas show the release locations of particle tracking. Blue lines show tracer trajectories for each particle. The numbers above the figures represent the release dates of particles.

**Figure 5 animals-14-00941-f005:**
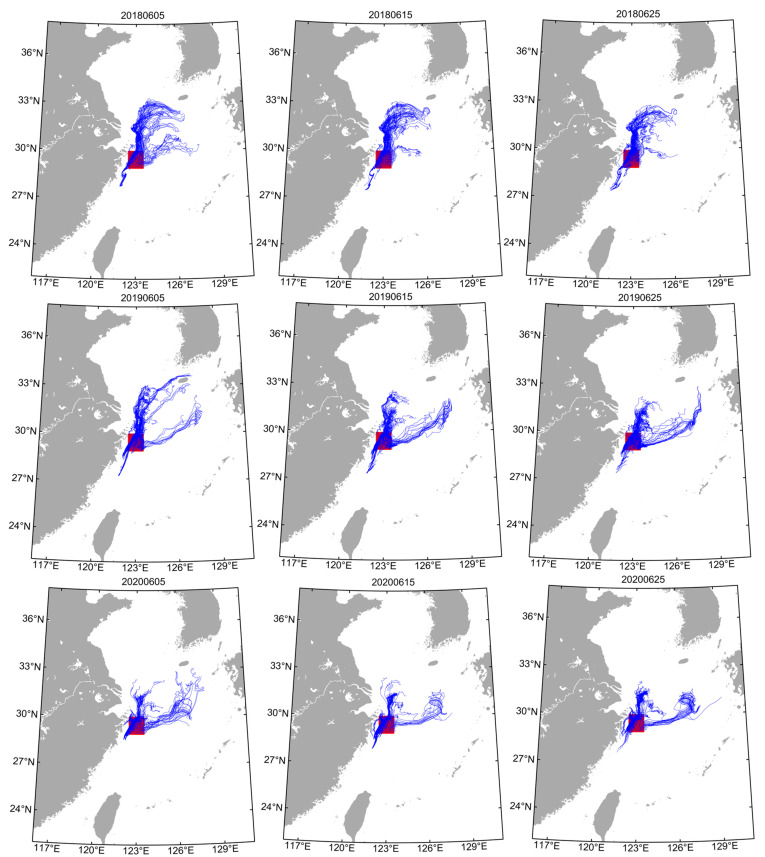
Particle tracking trajectories of the June group (summer stock) for 120 days in different sampling years. Red areas show the release locations of particle tracking. Blue lines show tracer trajectories for each particle. The numbers above the figures represent the release dates of particles.

**Figure 6 animals-14-00941-f006:**
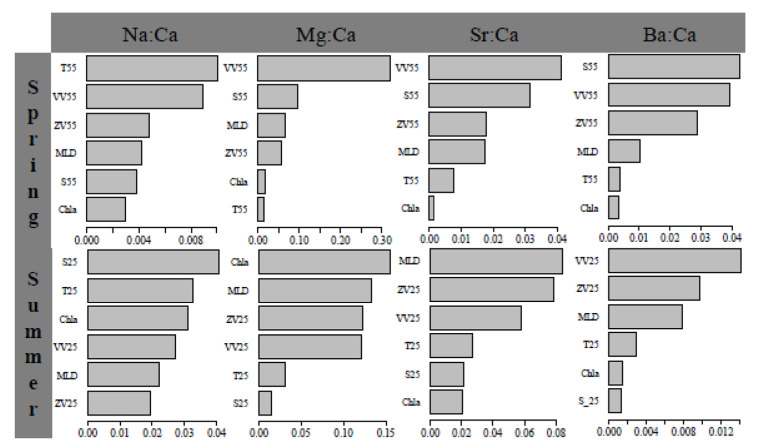
Importance weights of environmental variables in response to element ratios of spring stock and summer stock.

**Figure 7 animals-14-00941-f007:**
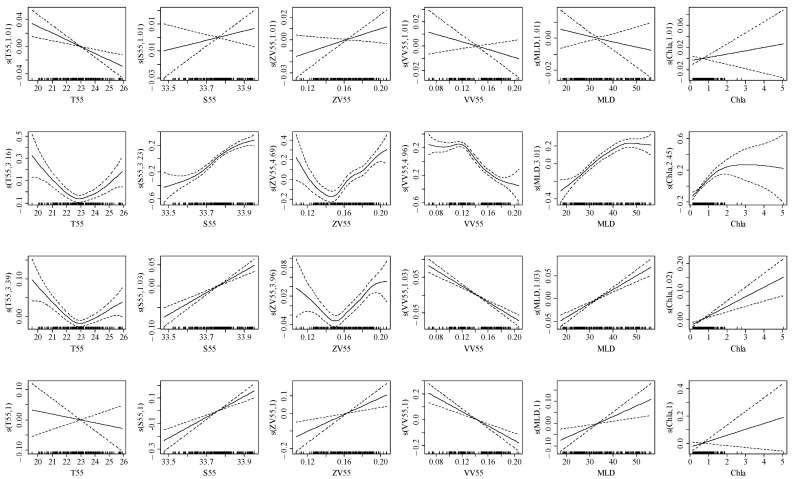
Effect of environmental variables on element ratios for spring stock based on GAMMs at early growth stages (within 360 μm, 120 d). The first row (six figures) to the fourth row show the effects of environmental variables on Na:Ca, Mg:Ca, Sr:Ca, and Ba:Ca, respectively. The solid line represents the fitted curve, and the dotted line represents the 95% confidence interval.

**Figure 8 animals-14-00941-f008:**
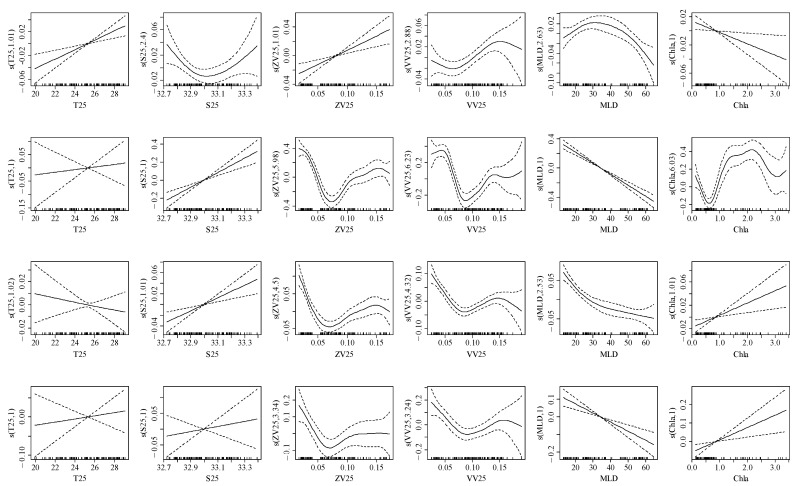
Effect of environmental variables on element ratios for the summer stock based on GAMMs at early growth stages (within 360 μm, 120 d). The first row (six figures) to the fourth row showed the effects of environmental variables on Na:Ca, Mg:Ca, Sr:Ca, and Ba:Ca. The solid line represents the fitted curve, and the dotted line represents the 95% confidence interval.

**Table 1 animals-14-00941-t001:** Information of statolith samples of *U. edulis* tested by micro-chemical analysis. Spring stock and summer stock hatched in March–May and June–July in this study, respectively. The sampling years are represented by 2018 (September 2018 to January 2019), 2019 (September 2019 to January 2020), and 2020 (September 2020 to December 2020).

Stock	Sampling Years	Quantity	Mantle Length/mm	Body Weight/g	Age/d	Maturity
Spring	2018	17	116–204	68–304	164–236	II–IV
	2019	13	128–284	70–356	182–301	II–IV
	2020	14	123–210	81–274	174–247	II–IV
Summer	2018	9	60–151	52–143	150–212	II–IV
	2019	8	68–149	43–137	153–213	II–IV
	2020	8	73–166	51–152	143–204	II–IV

**Table 2 animals-14-00941-t002:** Effect results of the relationship between element ratios and environment variables based on generalized additive mixed models for spring stock at early growth stages (within 360 μm, 120 d). Model significance: ns, *p* > 0.5; *, *p* < 0.05; **, *p* < 0.001; and ***, *p* < 0.0001. The environment variables in bold are the highest goodness-of-fit (top three) based on GAMMs.

Element Ratios	Environment Variables	Random Effects	Fixed Effects	Smooth Terms			Model Statistic	
		fYear SD	Std.Error	edf	F	*p*	BIC	Deviance Explained/%
Na:Ca	**T55**	**1.07 × 10^−11^**	**3.60 × 10^−3^**	**1.01**	**12.18**	*******	**−808.87**	**23.49**
	S55	9.05 × 10^−11^	3.68 × 10^−3^	1.01	0.99	ns	−798.09	7.23
	**ZV55**	**4.04 × 10^−11^**	**3.69 × 10^−3^**	**1.01**	**2.48**	*****	**−799.54**	**16.49**
	**VV55**	**4.51 × 10^−11^**	**3.70 × 10^−3^**	**1.01**	**1.66**	*****	**−798.81**	**12.25**
	MLD	1.96 × 10^−11^	3.72 × 10^−3^	1.01	0.79	ns	−797.93	6.03
	Chla	2.11 × 10^−11^	3.61 × 10^−3^	1.01	0.74	ns	−797.83	4.01
Mg:Ca	T55	5.27 × 10^−5^	7.09 × 10^−2^	3.16	8.83	***	123.76	7.53
	**S55**	**9.46 × 10^−2^**	**7.70 × 10^−2^**	**3.23**	**48.54**	*******	**16.61**	**26.90**
	ZV55	5.12 × 10^−5^	1.14 × 10^−1^	4.69	18.55	***	80.02	22.10
	**VV55**	**3.90 × 10^−2^**	**1.27 × 10^−1^**	**4.96**	**66.43**	*******	**−75.44**	**53.50**
	**MLD**	**4.68 × 10^−2^**	**7.32 × 10^−2^**	**3.01**	**47.59**	*******	**33.41**	**31.10**
	Chla	6.81 × 10^−2^	3.82 × 10^−2^	2.45	21.03	***	103.25	9.43
Sr:Ca	T55	1.04 × 10^−2^	2.09 × 10^−2^	3.39	9.76	***	−736.96	9.97
	**S55**	**2.39 × 10^−11^**	**4.09 × 10^−3^**	**1.03**	**44.27**	*******	**−750.91**	**12.60**
	ZV55	1.99 × 10^−2^	2.71 × 10^−2^	3.96	11.10	***	−733.88	8.50
	**VV55**	**2.66 × 10^−11^**	**3.93 × 10^−3^**	**1.03**	**79.48**	*******	**−779.68**	**20.50**
	**MLD**	**1.03 × 10^−10^**	**4.01 × 10^−3^**	**1.03**	**58.71**	*******	**−762.68**	**15.90**
	Chla	1.14 × 10^−11^	3.89 × 10^−3^	1.02	21.28	***	−729.44	6.06
Ba:Ca	T55	3.27 × 10^−7^	1.58 × 10^−2^	1.00	0.56	ns	109.60	7.54
	**S55**	**6.74 × 10^−9^**	**1.53 × 10^−2^**	**1.00**	**31.00**	*******	**80.71**	**19.12**
	**ZV55**	**3.97 × 10^−10^**	**1.57 × 10^−2^**	**1.00**	**10.27**	******	**100.80**	**12.92**
	**VV55**	**3.95 × 10^−2^**	**1.50 × 10^−2^**	**1.00**	**30.76**	*******	**82.76**	**17.86**
	MLD	3.53 × 10^−6^	1.51 × 10^−2^	1.00	9.02	**	102.00	9.50
	Chla	1.09 × 10^−7^	1.47 × 10^−2^	1.00	2.33	ns	107.99	5.42

**Table 3 animals-14-00941-t003:** Effect results of the relationship between element ratios and environment variables based on generalized additive mixed models for summer stock at early growth stages (within 360 μm, 120 d). Model significance: ns, *p* > 0.5; **, *p* < 0.001; and ***, *p* < 0.0001. The environment variables in bold are the highest goodness-of-fit (top three) based on GAMMs.

Element Ratios	Environment Variables	Random Effects	Fixed Effects	Smooth Terms			Model Statistic	
		fYear SD	Std.Error	edf	F	*p*	BIC	Deviance Explained/%
Na:Ca	**T25**	**9.15 × 10^−7^**	**5.74 × 10^−3^**	**1.01**	**12.25**	*******	**−334.69**	**18.62**
	**S25**	**2.99 × 10^−2^**	**1.83 × 10^−2^**	**2.40**	**4.05**	******	**−332.98**	**29.20**
	ZV25	3.47 × 10^−2^	4.77 × 10^−3^	1.01	12.65	***	−303.66	10.91
	VV25	3.63 × 10^−2^	2.37 × 10^−2^	2.89	5.38	**	−335.88	12.06
	MLD	3.41 × 10^−2^	2.41 × 10^−2^	2.63	7.76	**	−312.66	12.11
	**Chla**	**3.14 × 10^−2^**	**5.01 × 10^−3^**	**1.00**	**5.86**	*******	**−342.66**	**28.62**
Mg:Ca	T25	2.24 × 10^−10^	2.89 × 10^−2^	1.00	0.18	ns	84.54	10.71
	S25	2.46 ×10^−10^	2.50 × 10^−2^	1.00	26.30	***	63.10	17.20
	**ZV25**	**1.56 ×10^−1^**	**2.26 × 10^−1^**	**5.98**	**34.85**	*******	**1.55**	**56.10**
	VV25	1.11 ×10^−1^	2.49 × 10^−1^	6.24	22.04	***	27.82	49.10
	**MLD**	**3.14 × 10^−12^**	**2.30 × 10^−2^**	**1.00**	**109.70**	*******	**21.07**	**50.90**
	**Chla**	**1.48 × 10^−5^**	**2.74 × 10^−1^**	**6.03**	**20.99**	*******	**5.33**	**49.20**
Sr:Ca	T25	2.23 × 10^−11^	5.94 × 10^−3^	1.02	0.58	ns	−299.11	3.38
	S25	1.99 × 10^−11^	5.59 × 10^−3^	1.01	12.18	***	−310.00	8.52
	**ZV25**	**1.03 × 10^−6^**	**4.22 × 10^−2^**	**4.50**	**17.80**	*******	**−345.06**	**39.70**
	**VV25**	**7.32 × 10^−7^**	**3.85 × 10^−2^**	**4.32**	**16.20**	*******	**−340.04**	**36.20**
	**MLD**	**1.54 × 10^−16^**	**3.38 × 10^−2^**	**2.53**	**27.79**	*******	**−345.63**	**36.10**
	Chla	1.10 × 10^−11^	5.48 × 10^−3^	1.01	8.44	**	−306.46	5.70
Ba:Ca	T25	4.73 × 10^−7^	1.92 × 10^−2^	1.00	0.29	ns	−12.93	10.67
	S25	6.98 × 10^−7^	1.91 × 10^−2^	1.00	0.45	ns	−13.08	10.47
	**ZV25**	**3.83 × 10^−6^**	**1.10 × 10^−1^**	**3.34**	**5.40**	*******	**−21.52**	**23.90**
	**VV25**	**4.70 × 10^−6^**	**1.01 × 10^−1^**	**3.24**	**6.62**	*******	**−23.31**	**24.90**
	**MLD**	**3.17 × 10^−6^**	**1.83 × 10^−2^**	**1.00**	**20.58**	*******	**−31.47**	**24.60**
	Chla	2.32 × 10^−6^	1.71 × 10^−2^	1.00	8.61	**	−20.12	15.56

## Data Availability

The datasets analyzed during this current study are available from the corresponding author upon reasonable request.

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
