# Peer review of "Early Transport Patterns and Influencing Factors of Different Stocks of Uroteuthis edulis in the East China Sea"

_animals, 2024, doi:10.3390/ani14060941_

Round 1

Reviewer 1 Report

Comments and Suggestions for Authors

animals-2831731 Early Transport Patterns and Influencing Factors of Different Stocks of Uroteuthis edulis in the East China Sea”.

GENERAL COMMENT:

The work entitled Early Transport Patterns and Influencing Factors of Different Stocks of Uroteuthis edulis in the East China Sea is a good work.

 The work provides a scientific basis for a comprehensive understanding of the migration characteristics of U. edulis with different stocks. In this study particle tracing was used to simulate the early transport trajectories (within 120 days). Gradient Forest Method (GFM) and Generalized Additive Mixed Models (GAMMs) were used to analyze the key environmental variables that affect the statolith trace element ratio and relationships between trace ratio and environmental variables.

The subject of the study is interesting.

Central argument is supported by evidence and analysis.

The methodology described by the author is accurate.

This work needs some minor changes, for this reason I require minor revision.

DETAILED COMMENT:

·         Title

-The title is adequate.

·         Abstract

-The abstract is not well structured and the objective of the study is not clearly described. I suggest improving it.

·         Keywords are adequate

·         Introduction

The introduction section is not exhaustive. It must be improved.

·         Materials and Methods

The section is well written.

·         Results

This section is detailed

·         Discussion

I suggest to expand the discussion section.

Tables and Figures

Tables and Figures are clear and understandable.

·         References

The references are adequate.

Comments on the Quality of English Language

Minor editing of English language required.

Author Response

GENERAL COMMENT:

The work entitled “Early Transport Patterns and Influencing Factors of Different Stocks of Uroteuthis edulis in the East China Sea” is a good work.

The work provides a scientific basis for a comprehensive understanding of the migration characteristics of U. edulis with different stocks. In this study particle tracing was used to simulate the early transport trajectories (within 120 days). Gradient Forest Method (GFM) and Generalized Additive Mixed Models (GAMMs) were used to analyze the key environmental variables that affect the statolith trace element ratio and relationships between trace ratio and environmental variables.

The subject of the study is interesting.

Central argument is supported by evidence and analysis.

The methodology described by the author is accurate.

This work needs some minor changes, for this reason I require minor revision.

DETAILED COMMENT:

Title

1、The title is adequate.

Our Responses:

Thanks to the reviewers for their comments.

Abstract

2、The abstract is not well structured and the objective of the study is not clearly described. I suggest improving it.

Our Responses:

Thanks to the reviewers for their comments. This study used particle tracing method to determine the early transport process of individuals. Based on the statolith trace element can represent the individual migration, GAMMs of migration and environmental were established to determine the impact of environmental factors on the migration process, so as to clarify the individual early transport process and influencing factors. Based on your suggestions, the content here has been modified. Please see the revised manuscript for details (Lines 33-36).

3、Keywords are adequate

Our Responses:

Thanks to the reviewers for their comments.

Introduction

4、The introduction section is not exhaustive. It must be improved.

Our Responses:

Thanks to the reviewers for their comments. The second paragraph has been improved, and the objective of this study is clearly clarified in the last paragraph based on your suggestions. Please see the revised manuscript for details (Lines 59-66, 103-109).

5、Materials and Methods part. The section is well written.

Our Responses:

Thanks to the reviewers for their comments.

6、Results part. This section is detailed

Our Responses:

Thanks to the reviewers for their comments.

7、Discussion part. I suggest to expand the discussion section.

Our Responses:

Thanks to the reviewers for their comments. Based on your suggestions, the sentences have been improved in the discussion section. Please see the revised manuscript for details (Lines 428-434).

Tables and Figures

8、Tables and Figures are clear and understandable.

Our Responses:

Thanks to the reviewers for their comments.

References

9、The references are adequate.

Our Responses:

Thanks to the reviewers for their comments.

Reviewer 2 Report

Comments and Suggestions for Authors

 Nan Li et al. Animals-2831731

I had the opportunity to read the paper titled: Early Transport Patterns and Influencing Factors of Different Stocks of Uroteuthis edulis in the East China Sea (ECS). This paper should be interesting in the sense that it should allow a better knowledge of the U. edulis dispersal process in the ECS and the factors influencing the dynamics of this process. However, in some paragraphs, despite not being a native speaker of English, rephrasing sentences is essential to improve the presentation of the article.

Below are listed a few comments to improve the text.

L36 “were transport”? Please check. See also L330, L451-452.

L59-60 “Environmental driving forces and food availability are important factors affecting population recruitment during early transport”. It’s confusing. Please rephrase.

L60 “Current and temperature affect population recruitment by regulating eggs abundance and paralarva growth rate.” This is unconvincing and could be improved.

L59-65 It’s unclear. Please rewrite.

L67 “Uroteuthis edulis (U. edulis)” = “Uroteuthis edulis

L103-112 This is confusing. Please rephrase and improve by clearly stating the specific objectives of your study.

L115-117 This is incomprehensible. Please rephrase.

L124-127 "Figure 1. Sampling stations of Uroteuthis edulis in the East China Sea. Red and blue represented autumn and winter sampling sites, respectively. Areas A and B showed the locations of particle tracking released from the spring and summer stock, respectively. Red and blue stations represent autumn and winter sampling, respectively." This is very confusing caption. Please rephrase. Legend in Fig. 1 for sampling year could be also improved.

L124 Uroteuthis edulis = U. edulis, from here until the conclusion, authors should abbreviate the genus name. see also L159.

L190-192 "The tracers release locations of spring stock (Figure 1, area A) and summer stock (Figure 1, area B) were 122 °E to 123 °E, 26 °N to 27 °N and 122.5 °E to 123.5 °E, 29 °N to 30 °N, respectively (Figure 1)." Authors should clearly explain why this difference in location of tracer release sites. See also L363-364 "... the differences in the selection of tracer release areas may affect the real migration process".

L160 Please make Table 1 easier to read by improving the presentation of dates and clarifying the “Quantity” section.

L230 Please use here "Gradient Forest Method (GFM)" for clarity.

L238 also GAMMs  = Generalized Additive Mixed Models (GAMMs).

L122 “From 2018 to 2020, 290, 224 and 259 samples were respectively collected (Table S1).” However, these sample sizes do not match with Table S1, see L610. Please also add weight values. Mean and range values could be more interesting. These values also required in Table 1.

L254 “March group, April group and May group (spring stock)”, Please improve this repeat of "group".

L255 (Figure 4, S3, and S4) = (Figures 4, S3, and S4). See also L258, L3.

L262 in Fig. 4 What does the number above the figures represent? Why present the April group here instead of the March group? Please explain in caption.

L265 see also in Fig. 5

L323 “respond quickly” = “responds quickly”.

L610 Please italicize the scientific name of Uroteuthis edulis?

L613 What is meant by "stato l ith"? Please check.

L613 Please italicize the scientific name of Uroteuthis edulis?

Comments on the Quality of English Language

See comments and suggestions for authors. 

Author Response

I had the opportunity to read the paper titled: Early Transport Patterns and Influencing Factors of Different Stocks of Uroteuthis edulis in the East China Sea (ECS). This paper should be interesting in the sense that it should allow a better knowledge of the U. edulis dispersal process in the ECS and the factors influencing the dynamics of this process. However, in some paragraphs, despite not being a native speaker of English, rephrasing sentences is essential to improve the presentation of the article.

Below are listed a few comments to improve the text.

1、L36 “were transport”? Please check. See also L330, L451-452.

Our Responses:

Thanks to the reviewers for their comments. The expression here has been modified to "were transported". Please see the revised manuscript for details (Lines 37, 322, 450-451).

2、L59-60 “Environmental driving forces and food availability are important factors affecting population recruitment during early transport”. It’s confusing. Please rephrase.

Our Responses:

Thanks to the reviewers for their comments. The content here has been modified. Please see the revised manuscript for details (Lines 59-66).

3、L60 “Current and temperature affect population recruitment by regulating eggs abundance and paralarva growth rate.” This is unconvincing and could be improved.

Our Responses:

Thanks to the reviewers for their comments. The content here has been modified. Please see the revised manuscript for details (Lines 59-66).

4、L59-65 It’s unclear. Please rewrite.

Our Responses:

Thanks to the reviewers for their comments. The content here has been rewritten. This paragraph is intended to express the impact of environmental factors (temperature, food availability, etc.) in the habitat on individual growth and survival status during the early transport process, and then clarify the significance of this study to explore individual early migration and its influencing factors. The content here has been modified. Please see the revised manuscript for details (Lines 59-66).

5、L67 “Uroteuthis edulis (U. edulis)” = “Uroteuthis edulis

Our Responses:

Thanks to the reviewers for their comments. The content here has been modified. Please see the revised manuscript for details (Lines 68).

6、L103-112 This is confusing. Please rephrase and improve by clearly stating the specific objectives of your study.

Our Responses:

Thanks to the reviewers for their comments. Based on your comments, this paragraph has been rewritten to make the objectives of this study clearer. Please see the revised manuscript for details (Lines 103-109).

7、L115-117 This is incomprehensible. Please rephrase.

Our Responses:

Thanks to the reviewers for their comments. The content here has been modified. According to the sampling date, this study identified the samples sampled from September 2018 to January 2019, September 2019 to January 2020, and September 2020 to December 2020 as 2018, 2019, and 2020 samples, respectively. Please see the revised manuscript for details (Lines 112-115).

8、L124-127 "Figure 1. Sampling stations of Uroteuthis edulis in the East China Sea. Red and blue represented autumn and winter sampling sites, respectively. Areas A and B showed the locations of particle tracking released from the spring and summer stock, respectively. Red and blue stations represent autumn and winter sampling, respectively." This is very confusing caption. Please rephrase. Legend in Fig. 1 for sampling year could be also improved.

Our Responses:

Thanks to the reviewers for their comments. Figure 1 has been modified. Please see the revised manuscript for details (Figure 1).

9、L124 Uroteuthis edulis = U. edulis, from here until the conclusion, authors should abbreviate the genus name. see also L159.

Our Responses:

Thanks to the reviewers for their comments. The content here has been modified. Please see the revised manuscript for details.

10、L190-192 "The tracers release locations of spring stock (Figure 1, area A) and summer stock (Figure 1, area B) were 122 °E to 123 °E, 26 °N to 27 °N and 122.5 °E to 123.5 °E, 29 °N to 30 °N, respectively (Figure 1)." Authors should clearly explain why this difference in location of tracer release sites. See also L363-364 "... the differences in the selection of tracer release areas may affect the real migration process".

Our Responses:

Thanks to the reviewers for their comments. According to Li et al., 2023, hatching water temperature of spring stock and summer stock of U. edulis is different. Also, the results show that there are differences in the spawning ground of spring stock and summer stock of U. edulis through probability model, and they hatch in the waters of northeastern Taiwan and Zhoushan, Zhejiang, respectively (Li et al., 2023). Therefore, this study conducted particle release in the above two areas to simulate the early transport process. Lines 354-356, we discuss the limitations of this method. For each stock (spring stock or summer stock), the transport trajectory is related to the location of its spawning grounds and seasonal variations in currents, so the choice of particle release area may affect the migratory route of individuals.

Li, N.; Han, P.W.; Wang, C.; Chen, X.J.; Fang, Z. Migration routes of the swordtip squid Uroteuthis edulis in the East China Sea determined based on the statolith trace element information. Hydrobiologia. 2023, 850, 861-880. https://doi.org/10.1007/s10750-022-05129-8.

11、L160 Please make Table 1 easier to read by improving the presentation of dates and clarifying the “Quantity” section.

Our Responses:

Thanks to the reviewers for their comments. This study conducted sampling in 2018 (September 2018 to January 2019), 2019 (September 2019 to January 2020), and 2020 (September 2020 to December 2020). Based on the number and biological information (mantle length, etc.) of samples of each month, statoliths were ground to read the age of the samples. The sample for reading the age information is shown in Table S1. Therefore, considering the cost of trace element determination and the representativeness of the sample, this study selected spring stock and summer stock according to the sampling years to determine trace elements. The sampling date was clearly expressed according to your suggestion, and the body weight range was added. Please see the revised manuscript for details (Table 1).

12、L230 Please use here "Gradient Forest Method (GFM)" for clarity.

Our Responses:

Thanks to the reviewers for their comments. The content here has been modified. Please see the revised manuscript for details (Lines 222-223).

13、L238 also GAMMs  = Generalized Additive Mixed Models (GAMMs).

Our Responses:

Thanks to the reviewers for their comments. The content here has been modified. Please see the revised manuscript for details (Lines 230-231)

14、L122 “From 2018 to 2020, 290, 224 and 259 samples were respectively collected (Table S1).” However, these sample sizes do not match with Table S1, see L610. Please also add weight values. Mean and range values could be more interesting. These values also required in Table 1.

Our Responses:

Thanks to the reviewers for their comments. This study conducted sampling in 2018 (September 2018 to January 2019), 2019 (September 2019 to January 2020), and 2020 (September 2020 to December 2020). Based on the number of samples and biological information (mantle length, etc.) of each month, statoliths were ground to read the age of the samples. The sample for reading the age information is shown in Table S1. The sample sizes have been verified in sampling years according to your suggestions (accurately), and the information on body weight has been added. Please see the revised manuscript for details (Tables 1 and S1).

15、L254 “March group, April group and May group (spring stock)”, Please improve this repeat of "group".

Our Responses:

Thanks to the reviewers for their comments. The content here has been modified. Please see the revised manuscript for details (Lines 248-249, 251-252).

16、L255 (Figure 4, S3, and S4) = (Figures 4, S3, and S4). See also L258, L3.

Our Responses:

Thanks to the reviewers for their comments. The content here has been modified. Please see the revised manuscript for details (Lines 249 and 252).

17、L262 in Fig. 4 What does the number above the figures represent? Why present the April group here instead of the March group? Please explain in caption.

Our Responses:

Thanks to the reviewers for their comments. The number above the figures represents the release date of particle. Explanations have been made in the figure according to your suggestions, and the particle trajectories of the March group are shown in the main text, and the particle trajectories of the April group and May group are shown in the supplementary. Please see the revised manuscript for details (Figure 4).

18、L265 see also in Fig. 5

Our Responses:

Thanks to the reviewers for their comments. The number above the figures represents the release date of particle. Explanations have been made in the figure according to your suggestions, and the particle trajectories of the June group are shown in the main text, and the particle trajectories of the July group are shown in the supplementary. Please see the revised manuscript for details (Figure 5).

19、L323 “respond quickly” = “responds quickly”.

Our Responses:

Thanks to the reviewers for their comments. The content here has been modified. Please see the revised manuscript for details (Lines 315).

20、L610 Please italicize the scientific name of Uroteuthis edulis?

Our Responses:

Thanks to the reviewers for their comments. The content here has been modified. Please see the revised manuscript for details (Lines 610, Table S1).

21、L613 What is meant by "stato l ith"? Please check.

Our Responses:

Thanks to the reviewers for their comments. The word has been modified to "statolith". Please see the revised manuscript for details (Lines 613, Table S2).

22、L613 Please italicize the scientific name of Uroteuthis edulis?

Our Responses:

Thanks to the reviewers for their comments. The content here has been modified. Please see the revised manuscript for details (Lines 613, Table S2).

Reviewer 3 Report

Comments and Suggestions for Authors

Methodology Clarity: The methods section could benefit from more detailed explanations of the statistical models and particle tracer experiments. Clarifying these aspects will help readers understand the processes and replicate the study if needed.

Data Interpretation: While the data analysis is thorough, the connection between the results and the broader implications for marine ecology could be strengthened. Discussing the results in the context of existing literature would enhance the study's relevance.

Literature Review Expansion: The review of existing research seems limited. Expanding this section to include more recent studies or contrasting viewpoints could provide a more comprehensive background for your study.

Structural Improvements: The organization of the paper might benefit from a more logical flow, particularly in connecting the methodology, results, and discussion sections.

Statistical Validation: Additional statistical validation, such as sensitivity analysis or cross-validation of models, could strengthen the study's findings.

Addressing Limitations: More discussion on the limitations of your study, including potential biases and the generalizability of the findings, would be beneficial.

Graphics and Tables: Consider enhancing the clarity and quality of graphical representations and tables for better comprehension.

Language and Grammar: The manuscript would benefit from a thorough language editing to correct grammatical errors and improve readability.

Author Response

1、Methodology Clarity: The methods section could benefit from more detailed explanations of the statistical models and particle tracer experiments. Clarifying these aspects will help readers understand the processes and replicate the study if needed.

Our Responses:

Thanks to the reviewers for their comments. Based on your suggestions, we have added specific versions of each package in the R language. For specific operation procedures, please refer to the guidance in the R package. The particle tracer experiments is programmed in Matlab. If necessary, we can provide you with the code for data analysis. Please see the revised manuscript for details (Lines 242-244).

2、Data Interpretation: While the data analysis is thorough, the connection between the results and the broader implications for marine ecology could be strengthened. Discussing the results in the context of existing literature would enhance the study's relevance.

Our Responses:

Thanks to the reviewers for their comments. The main purpose of this study is to explore the early transport process and potential influencing factors of U. edulis. Since the results do not involve the analysis of ecological indicators (stomach content, trophic level), no in-depth analysis from an ecological perspective was conducted in the discussion section. Future research will take your suggestions into account and look at a broader ecological perspective.

3、Literature Review Expansion: The review of existing research seems limited. Expanding this section to include more recent studies or contrasting viewpoints could provide a more comprehensive background for your study.

Our Responses:

Thanks to the reviewers for their comments. The second paragraph has been improved, and the objective of this study is clearly clarified in the last paragraph based on your suggestions. Please see the revised manuscript for details (Lines 59-66, 103-109).

4、Structural Improvements: The organization of the paper might benefit from a more logical flow, particularly in connecting the methodology, results, and discussion sections.

Our Responses:

Thanks to the reviewers for their comments. According to your suggestions, the sentences and corresponding information in the Materials and Methods, Results and Discussion sections have been appropriately modified. Please see the revised manuscript for details (marked in red).

5、Statistical Validation: Additional statistical validation, such as sensitivity analysis or cross-validation of models, could strengthen the study's findings.

Our Responses:

Thanks to the reviewers for their comments. This study used the particle tracer experiment to simulate the transport trajectories of U. edulis in order to verify the early habitat range inferred by probabilistic model (Reference 4, Li et al., 2023). In the future, on the basis of collecting different size samples of the entire life history, bioenergetics and ecological models will be combined to simulate the dynamic migration process of this species.

6、Addressing Limitations: More discussion on the limitations of your study, including potential biases and the generalizability of the findings, would be beneficial.

Our Responses:

Thanks to the reviewers for their comments. The limitations of the results have been supplemented and improved. Please see the revised manuscript for details (Lines 336-337, 354-356, 421-422).

7、Graphics and Tables: Consider enhancing the clarity and quality of graphical representations and tables for better comprehension.

Our Responses:

Thanks to the reviewers for their comments. The figures have been redrawn based on your suggestions, and the original figure has been uploaded to the editorial office. Please see the revised manuscript for details (Table and figure).

8、Language and Grammar: The manuscript would benefit from a thorough language editing to correct grammatical errors and improve readability.

Our Responses:

Thanks to the reviewers for their comments. The grammar and tense of the full text sentences have been appropriately modified based on your suggestions (polished). Please see the revised manuscript for details (marked in red).

Round 2

Reviewer 2 Report

Comments and Suggestions for Authors

I checked the revised version of the study N°: animals-2831731 by Nan Li et al., which has been considerably improved. I am generally satisfied with the amendments made by the authors who have adequately addressed all my previous comments. I must congratulate the authors for the effort done in reviewing the manuscript.

In my opinion, this manuscript is ACCEPTABLE for publication in animals.